# Maternal Fibroblast Growth Factor 21 Levels Decrease during Early Pregnancy in Normotensive Pregnant Women but Are Higher in Preeclamptic Women—A Longitudinal Study

**DOI:** 10.3390/cells11142251

**Published:** 2022-07-21

**Authors:** Julieth Daniela Buell-Acosta, Maria Fernanda Garces, Arturo José Parada-Baños, Edith Angel-Muller, Maria Carolina Paez, Javier Eslava-Schmalbach, Franklin Escobar-Cordoba, Sofia Alexandra Caminos-Cepeda, Ezequiel Lacunza, Justo P. Castaño, Rubén Nogueiras, Carlos Dieguez, Ariel Iván Ruiz-Parra, Jorge Eduardo Caminos

**Affiliations:** 1Department of Physiology, School of Medicine, Universidad Nacional de Colombia, Bogota 11001, Colombia; judacostabo@unal.edu.co (J.D.B.-A.); mfgarcesg@unal.edu.co (M.F.G.); 2Department of Obstetrics and Gynecology, School of Medicine, Universidad Nacional de Colombia, Bogota 11001, Colombia; ajparadab@unal.edu.co (A.J.P.-B.); eangelm@unal.edu.co (E.A.-M.); airuizp@unal.edu.co (A.I.R.-P.); 3Department of Public Health, School of Medicine, Universidad Nacional de Colombia, Bogota 11001, Colombia; mcpaezl@unal.edu.co; 4Department of Surgery, School of Medicine, Universidad Nacional de Colombia, Bogota 11001, Colombia; jheslavas@unal.edu.co; 5Department of Psychiatry, School of Medicine, Universidad Nacional de Colombia, Bogota 11001, Colombia; feescobarc@unal.edu.co; 6Fundación Sueño Vigilia Colombiana, Bogota 111211, Colombia; 7School of Medicine, Universidad Pompeu Fabra, 08002 Barcelona, Spain; soficaminos2002@gmail.com; 8Centro de Investigaciones Inmunológicas Básicas y Aplicadas (CINIBA), Facultad de Ciencias Médicas, Universidad Nacional de La Plata, La Plata 1900, Argentina; ezequiellacunza@hotmail.com; 9Maimonides Institute of Biomedical Research of Cordoba (IMIBIC), Reina Sofia University Hospital, 14004 Cordoba, Spain; justo@uco.es; 10CIBERObn-Physiopathology of Obesity and Nutrition, Intituto de Salud Carlos III, 28029 Madrid, Spain; ruben.nogueiras@usc.es (R.N.); carlos.dieguez@usc.es (C.D.); 11Department of Physiology (CIMUS), School of Medicine, Instituto de Investigaciones Sanitarias (IDIS), Universidad de Santiago de Compostela, 15782 Santiago de Compostela, Spain

**Keywords:** FGF-21, pregnancy, preeclampsia, menstrual cycle

## Abstract

(1) Background: Fibroblast growth factor 21 (FGF-21) is an endocrine factor involved in glucose and lipid metabolism that exerts pleiotropic effects. The aim of this study was to investigate the serum FGF-21 profile in healthy and mild preeclamptic pregnant women at each trimester of pregnancy; (2) Methods: Serum FGF-21 levels were determined by ELISA in a nested case-control study within a longitudinal cohort study that included healthy (*n* = 54) and mild preeclamptic (*n* = 20) pregnant women, women at three months after delivery (*n* = 20) and eumenorrheic women during the menstrual cycle (*n* = 20); (3) Results: FGF-21 levels were significantly lower in the mid-luteal phase compared to the early follicular phase of the menstrual cycle in eumenorrheic women (*p* < 0.01). Maternal levels of FGF-21 were significantly lower in the first and second trimesters and peaked during the third trimester in healthy pregnant women (*p* < 0.01). Serum levels of FGF-21 in healthy pregnant were significantly lower in the first and second trimester of pregnancy compared with the follicular phase of the menstrual cycle and postpartum (*p* < 0.01). Serum FGF-21 levels were significantly higher in preeclamptic compared to healthy pregnant women during pregnancy (*p* < 0.01); (4) Conclusions: These results suggest that a peak of FGF-21 towards the end of pregnancy in healthy pregnancy and higher levels in preeclamptic women might play a critical role that contributes to protecting against the negatives effects of high concentrations of non-esterified fatty acids (NEFA) and hypertensive disorder. Furthermore, FGF-21 might play an important role in reproductive function in healthy eumenorrheic women during the menstrual cycle.

## 1. Introduction

Fibroblast growth factor 21 (FGF-21) is an endocrine hormone mainly produced in the liver during fasting state that plays several critical roles in metabolic stressful situations with diverse biological functions and implications for glucose as well as lipid metabolism, in both mice and humans [1,2]. Additionally, hepatic peroxisome proliferator-activated receptor α (PPARα) up-regulates FGF-21 expression in response to acute regulation of nutrient starvation or energy stress, elevated levels of circulating non-esterified fatty acids (NEFA) and ketogenic diets, and it is strongly suppressed by refeeding [3,4,5,6]. Furthermore, FGF-21 stimulates lipolysis in adipose tissue, promotes hepatic fatty acid oxidation and ketogenesis and is capable of lowering hepatic and systemic lipid levels [3,4].

Under physiological conditions, FGF-21 enhances insulin sensitivity and promotes glucose uptake in adipose tissue during refeeding and overfeeding in order to maximize the energy replenishment and to protect against lipotoxicity [2]. Additionally, FGF-21 increases energy expenditure and reduces body weight [7,8]. Moreover, recent studies have shown that circulating levels of FGF-21 are strongly associated with insulin-resistant states such as obesity and type 2 diabetes and inversely associated with both whole-body (muscle) and hepatic insulin sensitivity [9]. Furthermore, Zhang et al., showed that circulating FGF-21 was lower in lean controls as compared with obese nondiabetic subjects and was positively correlated with adiposity and fasting insulin, while it was negatively correlated with HDL cholesterol [10].

On the other hand, several studies have demonstrated that FGF-21 function is not exclusively limited to the control of energy metabolism, but it can be implicated with multiple physiological functions, such as oxidative stress, angiogenesis and inflammatory processes [11]. Therefore, this endocrine factor may play an important role in mediating pleiotropic effects in numerous critical pathways for cellular, tissue and whole-body homeostasis [12,13].

Several studies have assessed FGF-21 levels in pregnant women and its association with gestational diabetes mellitus [14,15]. However, its involvement in preeclampsia at different stages of pregnancy remains largely unknown. Preeclampsia is a hypertensive disorder of pregnancy characterized by systemic chronic vascular endothelial injury with chronic immune dysfunction and inflammation with increased levels of pro-inflammatory cytokines and diminished immunomodulatory factors [16]. As preeclampsia may be associated with endothelial dysfunction, it is not surprising that it can lead to a number of short and long-term health complications, including chronic hypertension, cardiovascular complications and renal complications [17]. Previous cross-sectional studies have demonstrated significant changes in maternal FGF-21 concentrations in preeclamptic women compared to healthy normotensive pregnant women [18,19,20]. Although the precise mechanisms responsible for driving preeclampsia remain to be elucidated, the role of FGF-21 in the pathophysiology of preeclampsia should be considered. Therefore, FGF-21 is a good candidate to be studied.

The aim of the current study was to determine the maternal serum profile of FGF-21 levels during the menstrual cycle in non-pregnant women and in a nested case-control study design within a prospective cohort study that included mild preeclamptic women and healthy normotensive pregnant women at each trimester of pregnancy and age-matched eumenorrheic women during the early-follicular and mid-luteal phases of the menstrual cycle, as well as their association with metabolic, hormonal and anthropometric parameters.

## 2. Materials and Methods

### 2.1. Ethical Aspects

In accordance with the Declaration of Helsinki, the current experimental protocol has been reviewed and approved by the Institutional Ethics Committees and Institutional Review Boards of the School of Medicine, Universidad Nacional de Colombia (Reference number: 011–165-18, June 2018). Written informed consent was obtained from all participants prior to participating in these studies. We conducted this study at The Engativa Hospital in Bogota Colombia and the Department of Obstetrics and Gynecology, School of Medicine, Universidad Nacional de Colombia.

### 2.2. Study Population

In this nested case-control study design within a prospective cohort study, healthy normotensive pregnant women (*n* = 52), pregnant women diagnosed with mild preeclampsia (*n* = 20) and mothers after three months of delivery (*n* = 20) were included. Additionally, age-matched healthy eumenorrheic women with a regular menstrual cycle (length between 28 and 30 days) and followed during the early-follicular (low estrogen and progesterone levels) (day 5 ± 1 from the onset of bleeding) and mid-luteal (high estrogen and progesterone levels) (day 22 ± 1 from the onset of bleeding) phases of the menstrual cycle were recruited for this study (*n* = 20). Additionally, cases were women with late-onset preeclampsia, and controls were age-matched normotensive pregnant women. Pregnant women were randomly selected from the original cohort study (*n* = 465).

### 2.3. Clinical Evaluation

Pregnant women were recruited for the study at the first prenatal visit in their first trimester of pregnancy (11–13 weeks). Gestational age was calculated based on last menstrual period and ultrasonography findings. Additionally, women were followed during their pregnancy to up to three months after delivery. Patient data were collected through interviews, physical examinations, anthropometric measurements and laboratory evaluations during the follow-up care and during each stage of pregnancy and postpartum. Sociodemographic characteristics, pregnancy outcomes, mode of delivery and complications during pregnancy and delivery were collected. In addition, clinical data on outcomes and complications were reported, including gestational age, birth weight, Apgar scores and pregnancy complications such as gestational diabetes mellitus (GDM), pregnancy-induced hypertension and preeclampsia.

The diagnosis of preeclampsia was made according to the evidence-based recommendations of the American College of Obstetricians and Gynecologists (ACOG) [21,22]. They established preeclampsia without severe features with blood pressure ≥ 140/90 mmHg screened on two occasions at least 4 h apart after the 20th week of gestation (in a woman with a previously normal blood pressure), and proteinuria (≥300 mg per 24-h urine collection) and severe preeclampsia with blood pressure ≥ 160/110 mmHg on two occasions at least 4 h apart after 20 weeks of gestation in a woman with a previously normal blood pressure or with any of the following features: thrombocytopenia, renal insufficiency, impaired liver function, pulmonary edema, cerebral or visual symptoms. Furthermore, we determined serum levels of pro and anti-angiogenic factors involved in preeclampsia, soluble fms-like tyrosine kinase 1 (sFlt-1) and placental growth factor (PlGF) in healthy pregnant and preeclamptic women. Serum samples were measured using human ELISA kits for sFlt-1 (ab119613) and PlGF (ab100629) purchased from Abcam^®^, (Boston, MA, USA). After log transformation of data of serum levels of sFlt-1 and PlGF, the Log (sFlt-1)/Log (PlGF) ratio was determined. It is important to highlight that only pregnant women diagnosed with mild preeclampsia were included, due to the low prevalence of severe preeclampsia in the current cohort study.

The inclusion criteria for this study were: all women were healthy at the start of the study, aged 17 to 38 years at conception without pregnancy-associated hypertension at the inclusion time and previous favorable pregnancy outcomes. The exclusion criteria included preexisting diabetes mellitus (DM), gestational diabetes, preexisting chronic hypertension, and liver, kidney or cardiovascular disease.

### 2.4. Biochemical and Hormonal Analysis

At each prenatal care visit and postpartum, blood samples were collected for biochemical and hormonal analyses. Laboratory values were generated from venous blood samples of mothers taken at each follow-up visit in the morning (7:00–8:00) and after overnight fast of ≥8 h. Blood samples were centrifuged, and the serum was separated and then stored in aliquots at −70 °C. Glucose, triglycerides (TG), total cholesterol (TC) and high-density lipoprotein cholesterol (HDL-C) were determined by standard methods (SPINREACT, Santa Coloma, Spain). Serum levels of C-reactive protein (hsCRP) were analyzed with a high-sensitivity homogenous immunoassay (BS-400 Chemistry Analyzer, Mindray, Shenzhen, China). Maternal insulin levels were analyzed by electrochemiluminescence immunoassay (Roche Elecsys 1010 Immunoanalyzer Boulder, Indianapolis, IN, USA). The HOMA-IR calculation was performed as described previously: [fasting insulin (μUi/mL) × fasting glucose (mmol/L)]/22.5 [23]. Serum progesterone levels were measured in healthy non-pregnant women by immunoassay (Roche Elecsys 1010 Immunoanalyzer Boulder, Indianapolis, IN, USA).

Human FGF-21 were analyzed in the base line in serum samples from the different experimental groups using a commercial ELISA kit from R&D Systems, Inc. (DF2100-Minneapolis, MN, USA), following the instructions of the manufacturer. As reported by the manufacturer, the detection limit for the assay was 8.69 pg/mL. The intra-assay and inter-assay coefficient of variation were ≤3.5% and ≤5.2%, respectively. Duplicate measurements of serum FGF-21 were made in the same samples.

### 2.5. Statistical Analysis

All statistical analyses were performed using R-version 3.4.0 (Vienna, Austria). Shapiro-Wilk’s test was performed to examine the normality of the data. Normally distributed data were expressed as mean and standard deviation (SD) and differences were analyzed by a two-tailed Student’s *t*-test. Non-normally distributed data were expressed as median and interquartile ranges (interquartile range (IQR)) and compared using the Mann-Whitney test. Non-normal data were normalized by log transformation and comparisons made using an unpaired *t*-test. The longitudinal data were compared using analysis of variance (ANOVA) for repeated measures. Pearson’s correlation coefficient was used to examine the relationship between serum concentrations of FGF-21 and other variables of this study. The receiver operating characteristic (ROC) curve was used to analyze the predictive value of FGF-21 for preeclampsia, and analysis was performed to determine the optimal cut-off points from the curves. A *p*-value < 0.05 was considered statistically significant.

## 3. Results

### 3.1. Demographic and Clinical Characteristics

The demographic and clinical characteristics of the healthy pregnant women, mild preeclamptic women and healthy eumenorrheic women included in the study are shown in Table 1 and Table 2 and Appendix A, respectively. Comparison of characteristics between healthy pregnant women and mild preeclamptic women in first, second and third trimesters of pregnancy are shown in Appendix A. Table 2 shows the significant differences throughout the three trimesters of pregnancy between healthy and preeclamptic pregnant women in the variables body mass index (BMI), systolic blood pressure (SBP), diastolic blood pressure (DBP), medium blood pressure (MBP), serum insulin levels, serum leptin levels and serum levels of FGF-21.

Furthermore, it should be noted that no significant correlation between serum levels of FGF-21 of healthy pregnant women and healthy eumenorrheic women were found, neither with clinical, anthropometric, biochemical, hormonal variables and insulin resistance indices throughout gestation (Appendix A).

### 3.2. Serum Levels of Progesterone in Non-Pregnant Women

Serum progesterone levels were determined in non-pregnant women during the follicular (days 3–5) and mid-luteal (days 21–23) phases of the menstrual cycle. Serum progesterone concentrations were significantly lower during the follicular phase compared with the luteal phase of the menstrual cycle Appendix A. Additionally, this study found a significant negative correlation between serum levels of FGF-21 with serum levels of progesterone (r = −0.421122, *p* = 0.0085).

### 3.3. Serum Levels of FGF-21 in Eumenorrheic Women

As can be observed from Figure 1, serum levels of FGF-21 were significantly lower in the luteal phase of the menstrual cycle when compared with the follicular phase in healthy eumenorrheic women (*p* < 0.0031).

### 3.4. Serum Levels of FGF-21 in Healthy Pregnant Women

Serum levels of FGF-21 vary significantly throughout gestation in both healthy (*p* < 0.0000) and preeclamptic (0.0036) pregnant women (Figure 1, Appendix A). Furthermore, serum levels of FGF-21 are significantly lower in the first and second trimesters of gestation, reaching peak serum levels in the third trimester of pregnancy in both healthy (*p* < 0.0000) and preeclamptic pregnant women (*p* < 0.0036) (Figure 1, Appendix A).

Furthermore, serum levels of FGF-21 are significantly lower in the first and second trimesters of pregnancy when compared to the follicular phase of the menstrual cycle in eumenorrheic women (Figure 1 and Appendix A). Serum levels of FGF-21 did not show significant differences between the luteal phase of the menstrual cycle of healthy eumenorrheic women with the first and second trimesters of healthy pregnant women (Figure 1 and Appendix A).

### 3.5. Serum Levels of FGF-21 in Postpartum Period

Serum levels of FGF-21 were significantly higher in postpartum women when compared to the luteal phase of the menstrual cycle, and the first or second trimesters of healthy pregnant women (Figure 1 and Appendix A). Serum FGF-21 levels did not differ significantly between the follicular phase of the menstrual cycle in healthy eumenorrheic women, and the third trimester of pregnancy or the postpartum period (Figure 1, Appendix A).

### 3.6. Serum Levels of FGF-21 in Pregnant Women with Preeclampsia

Serum FGF-21 levels were significantly higher during pregnancy in preeclamptic women when compared with healthy pregnant women (1st: *p* < 0.0064; 2nd: *p* < 0.0164; and 3rd: *p* < 0.0046, trimesters of pregnancy) (Figure 1, Appendix A).

### 3.7. Area under the ROC Curve (AUC) for Serum FGF-21 Levels

The area under the receiver operating characteristic curve (AUC ROC) for predicting the development of the adverse maternal outcome of mild preeclampsia (dependent variable) from serum levels of FGF-21 levels (independent variable) was determined in the 1st (0.681 (95% confidence interval 0.537–0.826)), 2nd (0.644 (95% confidence interval 0.501–0.788)) and 3rd (0.680 (95% confidence interval 0.523–0.836)) trimesters of pregnancy (Figure 2a–c, respectively). Additionally, the optimal cut-off points of serum levels of FGF-21 for predicting adverse maternal outcome of mild preeclampsia during the 1st, 2nd and 3rd trimesters of pregnancy are shown in Figure 2a–c, respectively.

### 3.8. Evaluation of sFlt-1/PlGF Ratio

The Log (sFlt-1)/Log (PlGF) ratio was determined in healthy and preeclamptic pregnant women and compared between the two groups at each trimester of pregnancy as described in Table 3. The results showed that the Log(sFlt-1)/Log(PlGF) ratio was significantly different between healthy pregnant and preeclamptic women at each trimester of gestation (Table 3).

## 4. Discussion

This study demonstrated for the first time that circulating levels of FGF-21 change during the menstrual cycle in eumenorrheic women and showed that circulating levels of FGF-21 vary throughout the three trimesters of gestation in healthy pregnant women and women with mild pre-eclampsia.

We have described, for the first time, circulating levels of FGF-21 during the follicular and luteal phase of normal menstrual cycle. The statistical analysis shows that serum levels of progesterone are negatively correlated with serum levels of FGF-21 in healthy eumenorrheic females. In fact, circulating levels of FGF-21 are significantly lower during the mid-luteal phase (high progesterone levels) and significantly higher during the early follicular phase of the menstrual cycle (low progesterone levels). These results agree with other studies where serum levels of FGF-21 showed a significant negative correlation with progesterone levels in menopausal and pre-menopausal women [24]. Therefore, the profiles of progesterone could participate in the regulation of circulating levels of FGF-21 during the menstrual cycle.

On the other hand, FGF-21 plays an important role in regulating lipid and glucose metabolism and energy balance [25,26]. Recently, Zhang et al., demonstrated that circulating FGF-21 levels were positively correlated with adiposity, fasting insulin and triglyceride levels and demonstrated a negative association with HDL-cholesterol levels [10]. Therefore, circulating FGF-21 levels were significantly higher in nondiabetic subjects with obesity when compared to lean healthy control subjects, indicating a potential resistance to FGF-21 in these subjects [10]. Human pregnancy is characterized by marked changes in maternal lipid metabolism. It could be divided into an early anabolic phase, which occurs in the first two trimesters of gestation and is characterized by the increase in maternal fat depots facilitated by insulin, and a later catabolic phase, which occurs during the third trimester and is characterized by increased maternal adipose tissue breakdown, hypertriglyceridemia as a result of insulin resistance and estrogen and other placental hormonal effects [27,28]. Thus, the present study shows that towards the end of pregnancy, circulating FGF-21 levels increased significantly in accordance with previous studies of exacerbated hypertriglyceridemia and insulin resistance condition [10,27,28,29], demonstrating a rise in circulating FGF-21 levels during late pregnancy due to catabolic state.

Preeclampsia is a pregnancy hypertensive disorder whose etiology remains obscure, but several metabolic and predisposing maternal conditions substantially increase the risk of late-onset preeclampsia, often associated with insulin resistance, high BMI and excessive weight gain during pregnancy [30,31,32,33]. On the other hand, it has been suggested that early-onset preeclampsia is most strongly associated with placental factors and may have a more severe disease course than late-onset preeclampsia [30,31,32,33]. It is known that preeclampsia as a complication of pregnancy is characterized by chronic conditions of endothelial damage, high production of ROS, suppression of endothelial NO synthesis and dysregulation in vascular tone modulation [34].

Several studies have shown that NEFA concentrations are significantly increased in women with pre-eclampsia compared with healthy pregnant women throughout pregnancy [35,36,37]. A recent in vitro study has demonstrated that FGF-21 administration to cerebral microvascular endothelial cells (CMEC) upon hypoxia stress inhibited the hypoxia-induced apoptosis and oxidative stress in cultured cells [38]. The present study demonstrated significantly higher levels of FGF-21 during late pregnancy in healthy pregnant women and in preeclamptic women. Nevertheless, the levels of FGF-21 were significantly higher in preeclamptic women compared to healthy pregnancies in each trimester of gestation. Thus, it is possible that elevated levels of NEFA in preeclamptic women compared with healthy pregnant women throughout pregnancy might contribute to the rise in circulating levels of FGF-21. Additionally, these findings suggest that high levels of FGF-21 in the third trimester in women with mild preeclampsia, might contribute to protection against injury associated with high levels of NEFA and hypertensive disorders of pregnancy.

We had previously determined the serum levels of NEFA by capillary gas chromatography-mass spectrometry (GC-MS) in a longitudinal cohort study of healthy pregnant women and three months postpartum and in non-pregnant women [39]. The results demonstrated that total levels of NEFA increased significantly during the mid-to-later pregnancy and similar results were reported by Herrera et al. [29]. It is important to highlight that the pregnant women were randomly selected from the same original cohort of a NEFA study [39]. Additionally, previous studies have shown that circulating NEFA were significantly higher in women with preeclampsia than in normotensive pregnancies [37,40].

Several experimental in vitro and in vivo studies have shown that elevated levels of NEFA may lead to an increase in β-oxidation of fatty acids, resulting in increased reactive oxygen species (ROS) production [41,42]. This ROS accumulation also induces endoplasmic reticulum stress and cell apoptosis, and promotes endothelial dysfunction associated with reduced nitric oxide (NO) bioavailability and progression of inflammatory disorders by high levels of pro-inflammatory cytokines produced predominantly by activated macrophages [43,44]. In contrast, recent studies have shown that FGF-21 exerts protective effects via attenuation of oxidative stress mediated by ROS, inflammation, vascular protective activities, lipotoxicity mainly associated with dysfunctional signaling in insulin resistance, and apoptosis [45,46,47]. Additionally, previous studies have demonstrated that high levels of NEFA are involved in triggering hepatic FGF-21 production, and conditions with high insulin resistance, such as obesity, are associated with FGF-21-resistant states, and thus showing increased circulating levels of endogenous FGF-21 [10,48]. NEFA changes during pregnancy reached the highest levels in the third trimester and were positively associated with maternal insulin resistance [39,49,50,51]. Another effect of high free fatty acid (FFA) levels is to increase levels of ROS in endothelial cells and smooth muscle cells. They can then react with NO, which can induce abnormal activation of the renin-angiotensin system [51]. In the present study, at the end of pregnancy when the switch to net catabolic state occurs, the peak of serum FGF-21 concentration coincides with higher NEFA levels in healthy and preeclamptic pregnant women.

Peroxisome proliferator-activated receptor α (PPARα), is a ligand-activated transcription factor associated with transcriptional regulation of key target genes involved in lipid and glucose metabolism and expressed mainly in tissues with elevated capacity for fatty acid oxidation, such as liver, heart and skeletal muscle [52,53]. Moreover, recent studies have revealed that PPARα is activated by different ligands, among which are the fatty acids that control the expression of genes involved in lipid metabolism [52,53]. A recent in vitro study of hepatocyte cells demonstrated that NEFA stimulates FGF-21 expression and secretion, and, as noted previously, liver PPARα is involved in hepatic lipid metabolism and thus may play a role in FGF-21 expression [6]. Therefore, the activation of PPARα by NEFA up-regulates FGF-21 expression and secretion during fasting, starvation and ketogenic diets, promoting lipolysis in adipose tissues and leading to increased hepatic fatty acid oxidation and ketogenesis [2,3]. Moreover, a recent study showed that circulating plasma levels of FGF-21 correlated directly with hepatic and whole-body (muscle) insulin resistance [9]. In the current study, we have shown that circulating FGF-21 levels have responsiveness changes during pregnancy and are likely driven, at least in part, by placental hormones over the course of pregnancy. In a human longitudinal physiological study, Powe et al., using hyperinsulinemic-euglycemic clamp techniques, demonstrated that there was a significant increase in insulin sensitivity from pre-pregnancy to early pregnancy and a significant decrease from early pregnancy to late pregnancy [54]. Thus, these results regarding insulin resistance correlate in the same way as the circulating FGF-21 levels throughout gestation described in the present study. To support the idea that the regulatory mechanism may link FGF-21 and NEFA during pregnancy, more in vitro and in vivo studies are necessary.

Angiotensin II is a critical factor of the renin-angiotensin system and plays an important role in hypertension and renal failure through influences on oxidative stress, endoplasmic reticulum stress, inflammation and transcription factor activation [55]. Additionally, different studies have shown that Angiotensin II is involved in ROS generation in the pathogenesis of hypertension [56]. As mentioned above, high levels of FFA directly affect the transcription of multiple genes associated with inflammation and oxidative stress in the endothelium [57]. Recently, Xuebo et al., demonstrated in knockout FGF-21 mice significant angiotensin II-induced high blood pressure and vascular dysfunction, whereas replenishment with FGF-21 reversed the hypertension impairment [58]. Recent cross-sectional studies showed that maternal serum levels of FGF-21 were higher in preeclamptic pregnant women when compared with normotensive control groups [18,20]. Furthermore, NEFA and triglycerides were increased in preeclamptic patients compared with the control group [59]. In accordance with the results obtained by Lin Jiang et al., the current study presents for the first time the longitudinal profile of FGF-21 in healthy and preeclamptic women, where FGF-21 levels are significantly elevated at each trimester of pregnancy in preeclamptic women compared with normotensive women, results that could correlate FGF-21, hypertension and NEFA during pregnancy [20].

A recent analysis of the human tissue-specific expression study of different organs and tissues using next-generation sequencing by Fagerberg et al., demonstrated that FGF-21 is predominantly expressed in liver tissue [60]. Additionally, Nitert et al., demonstrated that placental FGF-21 mRNA and protein expression were similar in women with preeclampsia compared to normotensive control pregnant women [19]. On the other hand, previous studies have shown that circulating levels of FGF-21 are increased in pathologies such as obesity, metabolic syndrome, insulin resistance, cardiovascular disease and different chronic inflammatory processes [61]. Pregnancy, as a temporary diabetogenic state mainly caused by insulin resistance, has circulating concentration of FGF-21 due to hepatic origin; therefore, it is possible that its regulation mechanisms might be modulated by different hormones of placental origin. Further research is needed to support the hypothesis that placental hormones play a critical role in the liver FGF-21 regulation mechanism during pregnancy.

Finally, our results show that serum FGF-21 levels are significantly higher in preeclamptic pregnant women compared to healthy normotensive pregnant women in the three trimesters of pregnancy in a longitudinal cohort, and these findings are broadly consistent with other cross-sectional studies [18,20]. Moreover, the AUC ROC for predicting adverse maternal outcome of mild preeclampsia from serum FGF-21 levels was determined in the first (0.681 (95% confidence interval 0.537–0.826)) which was slightly better for prediction than during the second (0.644 (95% confidence interval 0.501–0.788)) and third (0.680 (95% confidence interval 0.523–0.836)) trimesters of pregnancy. The results of the present study demonstrate that circulating levels of FGF-21 differ significantly at each trimester of pregnancy between healthy and preeclamptic pregnant women. Similarly, the Log (sFlt-1)/Log (PlGF) ratio differs significantly between healthy pregnant and preeclamptic women at each trimester of gestation. Therefore, the cut-off points in the levels of FGF-21 obtained through the different ROC curves in each trimester of pregnancy could contribute to the risk prediction of preeclampsia and further studies are needed to confirm a relationship between FGF-1 and Log (sFlt-1)/Log (PlGF) ratio and the outcome of preeclampsia [62].

## 5. Conclusions

These results suggest that a peak of FGF-21 towards the end of pregnancy in healthy pregnancy women and higher levels of FGF-21 throughout gestation in preeclamptic women compared with healthy normotensive pregnant women might play a critical role that contributes to protecting against the deleterious effects of high NEFA concentrations and hypertensive disorder. Additionally, we suggested that progesterone levels might be the cause of lower FGF-21 serum levels during the mid-luteal phase compared with the early follicular phase of the menstrual cycle in eumenorrheic women and this hormone might have a significant physiological impact on sexual function.

## Figures and Tables

**Figure 1 cells-11-02251-f001:**
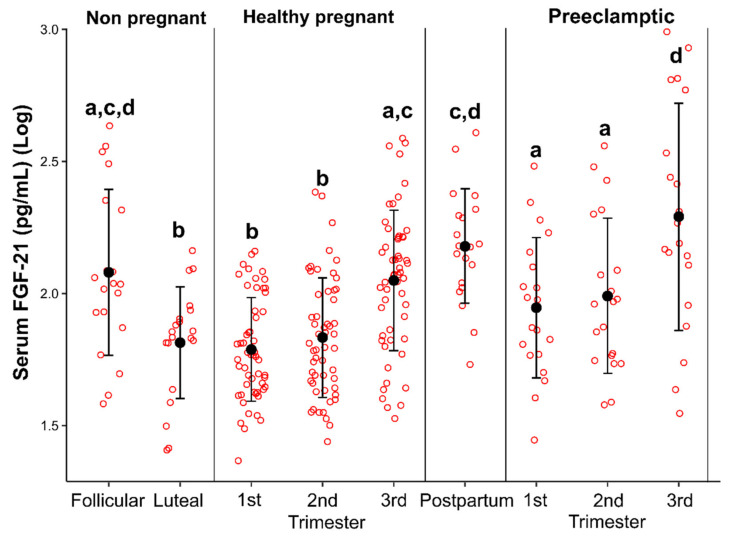
Dot plot diagram showing serum levels of FGF-21 in healthy eumenorrheic women during the early follicular and mid-luteal phases of the menstrual cycle, healthy pregnant women during pregnancy and postpartum, and women with preeclampsia. Different letters indicate statistically significant differences at *p* < 0.05.

**Figure 2 cells-11-02251-f002:**
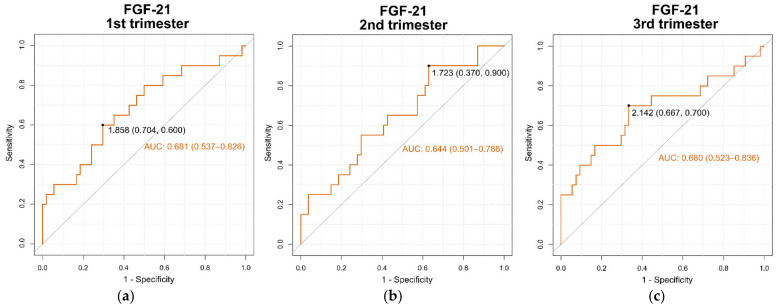
Receiver operating characteristic (ROC) curve of serum levels of FGF-21 (independent variable) measured in each trimester for the prediction of mild preeclampsia (dependent variable) during (**a**) 1st trimester of pregnancy; (**b**) 2nd trimester of pregnancy; (**c**) 3rd trimester of pregnancy. The optimal cut-off points (specificity, sensitivity) of serum FGF-21 levels in each trimester for predicting preeclampsia are shown. AUC: area under the curve (CI 95%).

**Table 1 cells-11-02251-t001:** Characteristics of healthy normotensive pregnant women during the 1st trimester, 2nd trimester and 3rd trimester of pregnancy.

Variables	1st Trimester	2nd Trimester	3rd Trimester	*p* Value (One-Way ANOVA Test)
Maternal Age (years)	25.6 ± 6.3 (17.0–38.0)	-	-	
Gestational Age (weeks)	12.13 ± 0.64 (11.0–13.6)	24.46 ± 0.69 (23.4–27.3)	354.7 ± 0.95 (33.4–38.6)	
BMI (kg/m^2^)	22.49 ± 2.34 (18.50–27.60)	24.31 ± 2.42 (19.20–30.20)	26.22 ± 2.49 (20.60–31.60)	0.0000
SBP (mmHg)	95.34 ± 8.69 (80.00–118.00)	92.97 ± 9.62 (70.00–111.00)	97.34 ± 9.46 (80.00–130.00)	0.0285
DBP (mmHg)	61.60± 6.28 (50.00–80.00)	60.45 ± 6.14 (50.00–90.00)	62.89 ± 8.51 (50.00–90.00)	0.1965
MBP (mmHg)	72.84 ± 6.33 (60.00–92.67)	72.29 ± 6.10 (56.67–94.00)	74.37 ± 7.96 (60.00–96.67)	0.0520
Blood Glucose (mg/dL)	77.91 ± 6.20 (64.00–92.00)	73.80 ± 5.27 (63.00–84.70)	74.15 ± 6.20 (63.00–88.00)	0.0001
Insulin (µUI/mL)	9.30 ± 4.25 (2.70–19.80)	11.12 ± 4.61 (3.20–24.10)	12.85 ± 5.82 (2.20–24.70)	0.0025
HOMA Index	1.78 ± 0.88 (0.45–4.01)	2.00 ± 0.83 (0.61–4.22)	2.44 ± 1.14 (0.76–4.52)	0.0266
Total Cholesterol (mg/dL)	166.39 ± 30.54 (111.02–251.60)	218.30 ± 39.43 (133.50–310.00)	246.12 ± 48.33 (152.20–362.80)	0.0000
HDL-C (mg/dL)	56.27 ± 10.01 (38.00–80.00)	67.39 ± 12.01 (42.97–90.27)	66.51 ± 11.94 (39.94–93.37)	0.0000
LDL (mg/dL)	116.96 ± 32.99 (53.70–197.12)	145.12 ± 45.53 (72.22–260.60)	157.60 ± 44.47 (75.00–257.97)	0.0000
VLDL (mg/dL)	22.10 ± 6.78 (10.56–42.06)	35.50 ± 10.26 (16.80–62.94)	48.64 ± 14.58 (21.40–83.04)	0.0000
Triglycerides (mg/dL)	116.65 ± 37.84 (69.80– 68.40)	183.42 ± 60.04 (84.00–380.90)	246.81 ± 77.26 (107.00–459.20)	0.0000
C-Reactive Protein	5.26 ± 2.98 (0.60–15.22)	4.73 ± 2.39 (0.69–10.00)	5.40 ± 3.12 (0.90–13.80)	0.6550
Leptin (ng/mL)	20.41 ± 7.24 (7.27–51.89)	25.40 ± 11.86 (6.02–70.28)	32.87 ± 13.40 (9.96–67.32)	0.0000
FGF-21 (pg/mL)	67.86 ± 31.76 (23.27–144.57)	78.37 ± 46.67 (27.49–241.98)	133.97 ± 84.56 (33.62–387.0)	0.0000

One-way ANOVA test was used for comparisons of continuous log-transformed values. Abbreviations: BMI, body mass index; HDL-C, high-density lipoprotein cholesterol; LDL, low-density lipoprotein; VLDL, very low-density lipoprotein; SBP, systolic blood pressure; DBP, diastolic blood pressure; MBP, medium blood pressure; HOMA, homeostatic model assessment; FGF-21, fibroblast growth factor 21. A *p* value of <0.05 was considered as statistically significant.

**Table 2 cells-11-02251-t002:** Characteristics of preeclamptic pregnant women during the 1st trimester, 2nd trimester and 3rd trimester of pregnancy.

Variables	1st Trimester	2nd Trimester	3rd Trimester	*p* Value (One-Way ANOVA Test)
Age (years)	23.6 ± 5.3 (17.0–34.0)	-	-	
Gestational Age (weeks)	12.2 ± 0.70 (11.2–13.4)	24.4 ± 0.57 (24.0–26.0)	35.0 ± 0.86 (34.0–37.0)	
BMI (kg/m^2^)	24.44 ± 3.11 (20.30–31.20)	26.83 ± 3.17 (22.40–33.04)	29.74 ± 2.97 (24.30–35.22)	0.0000
SBP (mmHg)	103.57 ± 7.74 (90.00–120.00)	103.39 ± 8.64 (88.00–126.00)	108.74 ± 12.07 (90.00–145.00)	0.1262
DBP (mmHg)	65.48 ± 7.27 (50.00–80.00)	64.74 ± 7.28 (56.00–82.00)	66.13 ± 6.92 (58.00–80.00)	0.7895
MBP (mmHg)	78.18 ± 6.80 (63.33–93.33)	77.62 ± 6.92 (67.33–91.33)	80.33 ± 7.45 (70.00–101.67)	0.3892
Blood Glucose (mg/dL)	80.50 ± 6.38 (72.00–99.00)	76.65 ± 7.60 (65.00–91.00)	73.53 ± 6.57 (64.00–90.00)	0.0293
Insulin (µUI/mL)	12.06 ± 3.94 (4.00–24.50)	14.90 ± 4.48 (8.40–24.40)	15.23 ± 6.16 (4.10–32.50)	0.0732
HOMA Index	2.39 ± 0.82 (0.71–4.78)	2.94 ± 1.06 (1.59–5.80)	2.83 ± 1.27 (0.66–6.50)	0.7825
Total Cholesterol (mg/dL)	173.04 ± 27.83 (103.30–233.40)	218.44 ± 41.84 (156.00–355.00)	233.84 ± 46.26 (149.10–344.30)	0.0000
HDL-C (mg/dL)	53.24 ± 10.76 (38.73–81.91)	64.13 ± 14.36 (42.68–98.08)	61.76 ± 14.40 (42.05–99.35)	0.0207
LDL (mg/dL)	118.85 ± 38.41 (52.30–207.49)	145.46 ± 57.24 (78.87–329.50)	151.35 ± 67.40 (50.89–308.00)	0.0776
VLDL (mg/dL)	23.68 ± 9.55 (7.92–43.20)	35.33 ± 14.25 (5.90–69.90)	51.01 ± 19.87 (11.40–87.40)	0.0000
Triglycerides (mg/dL)	124.88 ± 44.89 (71.60–216.00)	197.02 ± 88.32 (106.70–474.00)	263.55 ± 89.45 (135.40–437.00)	0.0000
C-Reactive Protein	6.10 ± 3.79 (0.65–13.74)	7.43 ± 2.86 (1.96–11.13)	7.02 ± 3.24 (1.57–16.24)	0.1526
Leptin (ng/mL)	34.97 ± 12.57 (14.24–70.84)	68.62 ± 32.49 (22.78–136.75)	89.98 ± 42.16 (24.27–184.53)	0.0000
FGF-21 (pg/mL)	105.84 ± 66.32 (27.87–303.59)	119.65 ± 91.96 (18.78–362.20)	303.24 ± 279.325 (35.1524–1052.7)	0.0036

One-way ANOVA test was used for comparisons of continuous log-transformed values. Abbreviations: BMI, body mass index; HDL-C, high-density lipoprotein cholesterol; LDL, low-density lipoprotein; VLDL, very low-density lipoprotein; SBP, systolic blood pressure; DBP, diastolic blood; MBP, medium blood pressure; HOMA, homeostatic model assessment; FGF-21, fibroblast growth factor 21. A *p* value of <0.05 was considered as statistically significant.

**Table 3 cells-11-02251-t003:** Evaluation of the Log (sFlt-1)/Log (PlGF) ratio in healthy pregnant and preeclamptic women at each trimester of gestation.

	Log(sFlt-1)/Log(PlGF) Healthy Pregnant Women	Log(sFlt-1)/Log(PlGF) Preeclamptic Women	*p* Value *t*-Test
1st trimester	2.02488 ± 0.322863	1.809 ± 0.240961	0.01038
2nd trimester	1.31829 ± 0.104664	1.23789 ± 0.0747276	0.00391
3rd trimester	1.4239 ± 0.161476	1.58632 ± 0.330844	0.01284
*p* value (ANOVA test)	0.0000	0.0000	

sFlt-1/PlGF: soluble fms-like tyrosine kinase 1/proangiogenic factor, placental growth factor (PlGF). A *p* value of <0.05 was considered as statistically significant.

## Data Availability

This study did not report any additional data.

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
