# Peer review of "Maternal Fibroblast Growth Factor 21 Levels Decrease during Early Pregnancy in Normotensive Pregnant Women but Are Higher in Preeclamptic Women—A Longitudinal Study"

_cells, 2022, doi:10.3390/cells11142251_

Round 1

Reviewer 1 Report

Dear Authors,

In general, your study complements well studies on the search for marker molecules and their relationship with the pathogenesis of preeclampsia. I would like to note the originality of your work, since the level of FGF21 was determined in non-pregnant women, as well as during the trimesters of pregnancy, which has not been found before. The only manuscript I found was on validation of a novel panel of biomarkers for a test for preeclampsia  (https://doi.org/10.1016/j.jpba.2022.114729). I have some minor questions and additions below.

3. Results

3.1. Demographic and clinical characteristic.

In Table 2 does not deciphering of the PAD and PAM abbreviations. Under the table, the decoding of other abbreviations is indicated - DBP, Diastolic blood pressure at blood draw; MBP., Medium blood pressure. Please correct it.

Have you measured estrogen and progesterone levels in non-pregnant women during follicular and mid-luteal phase sampling? If yes, did you calculate the correlation between estrogen,  progesterone levels and FGF21 levels?

3.5. Area under the ROC curve (AUC) for serum FGF-21 levels

Please, indicate which characteristics you included in ROC-analysis as dependent variables (response variables), and which ones as independent (predictor) variables.

4. Discussion

Line 254-276. Please, describe briefly, as the Results section already describes this in detail.

Please, specify whether you have carried out the determination of NEFA, since I did not find this data in the tables, and you mention it in the text (In the present study, at the 301 end of pregnancy when the switch to net catabolic state occurs, the peak of serum FGF-21 302 concentration coincides with the higher NEFA levels in healthy and preeclamptic preg- 303 nant women)

Author Response

We would like to thank the referee once more for sparing the time to write so many detailed and useful comments. Please see the attachment and find our response in the tittle " Reviewer #1 "

Reviewer 2 Report

Dear authors
You have submitted an interesting manuscript describing the association between pre-eclampsia and FGF-21.
In the text, you refer several times to the association of FGF-21 and estrogen and progesterone levels. What I miss in the text is a description of the mechanism behind these associations and, of course, the proof that these hormones are in a causal relationship.
Among the results, you do not mention the determination of the ratio of s-FlT-1 to PlGF, which is the gold standard for defining preeclampsia, together with ultrasound measurement. In this way, we could also demonstrate the correlation of FGF-21 with this ratio.
In your discussion, you discuss the association of high FGF-21 and NEFA concentrations as reported in the literature. In your conclusion, you say that higher levels of FGF-21 might play a critical role that contributes to conferring protection against the deleterious effects of high NEFA concentrations. This claim could be proven by actual NEFA results.
In Table 2, PAD and PAM codes are not explained.

Author Response

We would like to thank the referee once more for sparing the time to write so many detailed and useful comments. Please see the attachment and find our response in the tittle " Reviewer #2 "

Reviewer 3 Report

Julieth, et al., reported that the maternal FGF21 levels were decreased during the early pregnancy and remain higher in preeclamptic condition.

Here is some of the comments to make the publication more useful.

1.    Since several researchers investigated that the role of FGF21 during the pregnancy and preeclampsia condition – novelty is missing

2.    As authors mentioned in the conclusion part of the abstract authors are not showing relationship between FGF21-NEFA-hypertension.

3.    The manuscript would be more useful if the experimental study was focused more on serum biomarkers for NEFA-Hypertension.

Author Response

We would like to thank the referee once more for sparing the time to write so many detailed and useful comments. Please see the attachment and find our response in the tittle " Reviewer #3 "

Round 2

Reviewer 2 Report

All the requirements set by the reviewer have been met. The text you have added meets the requirements.
I consider the text suitable for publication.